# SModelS v2.3: Enabling global likelihood analyses

Mohammad Mahdi Altakach[1*], Sabine Kraml[1†], Andre Lessa[2‡],
Sahana Narasimha[3∘], Timothée Pascal[1§] and Wolfgang Waltenberger[3,4¶]

**1** Laboratoire de Physique Subatomique et de Cosmologie (LPSC), Université Grenoble-Alpes, CNRS/IN2P3, 53 Avenue des Martyrs, F-38026 Grenoble, France
**2** Centro de Ciências Naturais e Humanas, Universidade Federal do ABC, Santo André, 09210-580 SP, Brazil
**3** Institut für Hochenergiephysik, Österreichische Akademie der Wissenschaften, Nikolsdorfer Gasse 18, A-1050 Wien, Austria
**4** University of Vienna, Faculty of Physics, Boltzmanngasse 5, A-1090 Wien, Austria

★ altakach@lpsc.in2p3.fr , † sabine.kraml@lpsc.in2p3.fr , ‡ andre.lessa@ufabc.edu.br ,
∘ sahana.narasimha@oeaw.ac.at , § timothee.pascal@lpsc.in2p3.fr , ¶ walten@hephy.oeaw.ac.at

## Abstract

We present version 2.3 of SModelS, a public tool for the fast reinterpretation of LHC searches for new physics on the basis of simplified-model results. The main new features are a database update with the latest available experimental results for full Run 2 luminosity, comprising in particular a large variety of electroweak-ino searches, and the ability to combine likelihoods from different analyses. This enables statistically more rigorous constraints and opens the way for global likelihood analyses for LHC searches. The physics impact is demonstrated for the electroweak-ino sector of the minimal supersymmetric standard model.

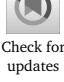

## Contents



# 1   Introduction

The lack of evidence for new physics in the LHC data puts stringent constraints on masses and couplings of new particles, predicted by theories beyond the standard model (BSM). However, searches for new physics at the LHC are made on a channel-by-channel basis in specific final states, and the results are typically presented in the context of simplified models. To obtain a comprehensive view of how the plethora of LHC results constrain new physics and which regions of parameter space remain viable –or perhaps even untested– it is therefore important to reinterpret the results of these searches in the context of realistic theoretical scenarios. This is particularly relevant for BSM theories which feature a large number of new particles with potentially very complex decay patterns, such as supersymmetric theories, including the minimal supersymmetric standard model (MSSM).

Reinterpretation [1–3] can be based on event simulation, reproducing as closely as possible the experimental analysis. Alternatively, it can be based on simplified-model results, ideally in the form of acceptance×efficiency values for all the regions considered in the analysis. This latter approach is the one taken in SMODELS [4–7]; it assumes that the kinematic distributions of the tested BSM scenario are to a good approximation the same as in the simplified model.

Independent of the approach, a crucial aspect of reinterpretation is the statistical modelling [8] in order to set a limit and/or derive a confidence level for the hypothesised signal. In recent years, the amount of information provided by experimental collaborations has increased significantly, allowing for a more robust statistical interpretation in phenomenological studies. This concerns in particular information on background correlations, in the form of covariance matrices [9] or complete statistical models [10], enabling the combination of disjoint signal regions (SRs) of an analysis in its reinterpretation [6, 11–14].

The next important step forward is the construction of global likelihoods from the combination of (approximately) orthogonal analyses. Such global approaches are attempted by, e.g., the GAMBIT collaboration [15] (see also [11, 14]) and the "protomodelling" project in [16]. The gain in exclusion power relative to single-analysis limits was recently demonstrated in [17] for models with varying degrees of complexity.

In this paper, we present the possibility of analyses combination in SMODELS v2.3. As mentioned, SMODELS is a public tool for the fast reinterpretation of LHC searches for new physics on the basis of simplified-model results. Its working principle is to decompose the signatures of full BSM scenarios into simplified-model components, which are then confronted against the experimental constraints from a large database of results. So far [7], constraints on a particular BSM point were considered purely on an analysis-by-analysis basis, i.e. for each experimental search separately. The new version presented in this paper now provides the possibility to combine likelihoods from different analyses, under the assumption that they are approximately uncorrelated. As we will show, this enables statistically more rigorous constraints and opens the way for global likelihood analyses with SMODELS. Besides the support of analysis combinations, the entire statistical computation has been refactored in v2.3. Moreover, support was added for simplified likelihoods in the formalism of [18] (SLv2, Gaussian

with a skew). The full list of improvements, including small bug fixes, can be found in the release notes.

The other significant news in version 2.3 is the database update with the latest available experimental results for full Run 2 luminosity, comprising in particular the full suit of (re)usable electroweak-ino (EW-ino) searches. Concretely, results from 9 ATLAS and 12 CMS publications were added to the database. With this, the SMODELS v2.3.0 database covers a total of 111 ATLAS and CMS searches (78 from Run 2 and 33 from Run 1); 17 of the ATLAS and 13 of the CMS searches are for full Run 2 luminosity. The physics impact of both, the database update and the new feature of analysis combination, is demonstrated by means of a case study of the EW-ino sector of the MSSM.

The paper is organised as follows. We begin in section 2 by reviewing the likelihood calculation in SMODELS. The combination of SRs within an analysis is explained in section 2.1, and the combination of likelihoods of orthogonal analyses in section 2.2. The new analyses in the database are presented in section 3, and the EW-ino case study in section 4. Section 5 contains a summary and conclusions. Four appendices complete the paper: Appendix A provides a complete list of the experimental results in the v2.3.0 database, Appendix B explains the new functionality of database add-ons, and Appendix C gives examples of combining more than two analyses. Finally, Appendix D contains auxiliary plots supplementing the results presented in section 4.

It is assumed that the reader is familiar with the concepts and usage of SMODELS. If this is not the case, we refer to [4–7] and the extensive online manual for a detailed introduction.

## 2 Likelihood calculation

SMODELS uses two types of experimental results: *-1-* upper limits (ULs) [4] on the cross sections of simplified models as function of the masses and, if relevant, the decay widths of the particles of the simplified model, and *-2-* efficiency maps (EMs) [5] in the same type of parameterisation. More specifically, EMs are acceptance×efficiency values for the various SRs of an experimental analysis. Whenever EMs are available, or have been produced by recasting like in [19], SMODELS can compute a likelihood for the assumed signal. This likelihood describes the plausibility of a signal strength $\mu$ given the data $D$:

$$\mathcal{L}(\mu, \theta|D) = P(D|\mu s + b + \theta)p(\theta). \tag{1}$$

Here, $s$ and $b$ are the number of predicted signal and background events, respectively, while $\theta$ denotes the nuisance parameters that describe the variations in the signal and background contributions due to systematic effects, with $p(\theta)$ being the probability distribution of the nuisances.

The simplest case, when `combineSRs=False` in the SMODELS parameters settings, is to compute the likelihood per SR. This assumes $p(\theta)$ to follow a Gaussian distribution centered around zero with a variance of $\delta^2$, whereas $P(D)$ corresponds to a counting variable and is thus described by a Poissonian. The likelihood for each SR thus takes the form [5]

$$\mathcal{L}(\mu, \theta|D) \propto \frac{(\mu s + b + \theta)^{n_{\text{obs}}} e^{-(\mu s + b + \theta)}}{n_{\text{obs}}!} \exp\left(-\frac{\theta^2}{2\delta^2}\right), \tag{2}$$

with $n_{\text{obs}}$ the number of observed events and $\delta$ the $1\sigma$ background uncertainty. Given the likelihood, a 95% confidence level limit on $\mu$, $\mu_{95}$, is computed using the $\text{CL}_s$ prescription [20], employing the test statistic $q_\mu$ according to Eq. (14) in [21]. SMODELS then reports the result for the most sensitive (a.k.a. "best") SR for each analysis.[1] Concretely, the standard output

---

[1] The most sensitive, or "best", SR is the one with the strongest expected limit.

consists of the expected and observed $r$-values, with $r$ defined as the ratio of the predicted fiducial cross section of the signal over the corresponding upper limit ($r \equiv 1/\mu_{95}$), as well as the values for the observed $\mathcal{L}_{\mathrm{BSM}} \equiv \mathcal{L}(\mu = 1)$, $\mathcal{L}_{\mathrm{SM}} \equiv \mathcal{L}(\mu = 0)$ and $\mathcal{L}_{\mathrm{max}} \equiv \mathcal{L}(\hat{\mu})$.

If information on the background correlations across SRs is provided by the experimental collaboration, either in form of a correlation or covariance matrix, or –better– in the form of a full statistical model, we can go a significant step further and compute the likelihood for the entire analysis, combining its SRs. To this end one has to set `combineSRs=True` in the SMODELS parameters settings. Three different approaches are now available in SMODELS as detailed below in section 2.1.

Moreover, independent of whether or not SRs are combined, SMODELS now offers the possibility to combine likelihoods from different analyses. This is described below in section 2.2 and constitutes the most important new feature of the package. Finally, the entire statistical computation has been refactored and centralized in v2.3 into the StatsComputer class.

## 2.1  Combination of signal regions within an analysis

**Simplified likelihood version 1: Gaussian uncertainties**

In this framework, initially introduced in [9] and available in SMODELS since v1.1.3 [6], all nuisance parameters are consolidated into a single distribution using a multivariate Gaussian. Concurrently, Poissonians are utilised to accommodate the statistical behaviour arising from counting events in individual signal regions. The likelihood for $N$ SRs takes the form

$$\mathcal{L}(\mu, \theta | D) \propto \prod_{i=1}^{N} \mathrm{Pois}\left(n_{\mathrm{obs}}^{i} | \mu s_i + b_i + \theta_i\right) \exp\left(-\frac{1}{2}\vec{\theta}^T V^{-1} \vec{\theta}\right), \tag{3}$$

where $\mu$ is the overall signal strength and $V$ represents the covariance matrix.[2] Signal uncertainties are neglected.

Referred to as SLv1 in this paper, this simplified likelihood approach holds the distinction of being the first technique that enabled the combination of signal regions in a non-trivial manner for phenomenologists whenever a correlation or covariance matrix is available for an analysis. The SLv1 has demonstrated satisfactory performance as long as the Gaussian approximation for the nuisances is valid. However, it may not be a good approximation in case of very small expected event yields.

**Simplified likelihood version 2: Gaussian with a skew**

A possible solution to account for non-Gaussian effects in the nuisances is incorporating a skewness term in the Gaussian distribution as proposed in [18]. In the formalism of [18], again assuming a Poisson statistics for the observed event counts, the likelihood takes the form

$$\mathcal{L}(\mu, \theta | D) \propto \prod_{i=1}^{N} \mathrm{Pois}\left(n_{\mathrm{obs}}^{i} | \mu s_i + \alpha_i + \beta_i \theta_i + \gamma_i \theta_i^2\right) \exp\left(-\frac{1}{2}\vec{\theta}^T \rho^{-1} \vec{\theta}\right), \tag{4}$$

which is referred to as SLv2 in this paper. Note that here the nuisances from eq. (3) have been reparametrised as $\theta_i \rightarrow \beta_i \theta_i$. The coefficients $\alpha_i$, $\beta_i$ and $\gamma_i$ can be related to the first three statistical moments. Specifically, the first moment is the mean, while the second moment is the covariance $V_{ij} = \beta_i \beta_j \rho_{ij} + 2\gamma_i \gamma_j \rho_{ij}^2$; the diagonal element of the third moment is $m_{3,i} = 6\beta_i^2 \gamma_i + 8\gamma_i^3$.[3] In the end, all that is effectively needed to extend the SLv1 to SLv2 is

---

[2]We recall that correlation matrix $\rho$ and covariance matrix $V$ are related by $V_{ij} = \rho_{ij}\delta_i\delta_j$.

[3]In the words of [22], $\alpha_i$ is the central value of the background prediction; $\beta_i$ corresponds to the effective sigma of the background uncertainty, with $\beta_i = \sqrt{V_{ii}}$ in the limit of symmetric uncertainties; and $\gamma_i$ describes the asymmetry of the background uncertainty.

the third moment $m_3$, which, given asymmetric background uncertainties $\delta_-$ and $\delta_+$, may be computed from a bifurcated Gaussian as [22]

$$m_3 = \frac{2}{\delta_- + \delta_+}\left[\delta_- \int_{-\infty}^{0} x^3 \, \text{No}\left(x; 0, \delta_-^2\right) dx + \delta_+ \int_{0}^{\infty} x^3 \, \text{No}\left(x; 0, \delta_+^2\right) dx\right], \qquad (5)$$

where No refers to the normal distribution. We note that the SLv2 has been technically available in SMODELS since a while, but was not used due to lack of experimental information. The CMS-SUS-20-004 analysis [23] (see section 3) is the first analysis to provide the required information. With this, SMODELS is the first reinterpretation tool to make actual use of the formalism of [18].

**HistFactory statistical models**

ATLAS searches are often based on HISTFACTORY [24] for their statistical modelling. Following [10], the collaboration has started to provide JSON serialisations of the full HISTFACTORY workspaces for results with full Run 2 luminosity (139/fb) on HEPDATA. Thus the full set of nuisance parameters, changes under systematic variations, and observed data counts are provided at the same fidelity as used in the experiment.

SMODELS supports the usage of these JSON-serialised statistical models since v1.2.4 [12] via an interface to the PYHF package [25], a pure-python implementation of the HISTFACTORY family of statistical models. This means that with `combineSRs=True`, whenever a HISTFACTORY statistical model is available in the database, the evaluation of the likelihood is relegated to PYHF (where, internally, the calculation is again based on the asymptotic formulas of [21]).

It has to be noted here that the evaluation of full HISTFACTORY models, which can have hundreds of nuisance parameters, can be very CPU intensive, in particular when combining analyses. For this reason, the official SMODELS database contains mostly simplified HISTFACTORY models [26], which were derived from the full ones by means of the SIMPLIFY [27] tool. The currently only exception is the ATLAS-SUSY-2019-08 analysis [28], for which the SIMPLIFY'ed statistical model does not reproduce the results from the full one well enough, and therefore the full one is kept as the default. In any case, when CPU performance is not an issue, the SIMPLIFY'ed statistical models in the database can be replaced by the full ones through a "full_llhds" database add-on as explained in Appendix B.

Some minor technical fixes have been implemented with respect to [12], without changing the code's core. The most noticeable changes include the suppression of the `datasetOrder` field from the `globalInfo.txt` file in the database entry, and the possibility to not remove the control regions from the HISTFACTORY statistical model by adding `includeCRs = True` in `globalInfo.txt`. The latter is used for two analyses so far: ATLAS-SUSY-2018-32 [29] and ATLAS-SUSY-2019-09 [30]. For all other analyses that make use of a HISTFACTORY statistical model, `includeCRs = False` by default.

## 2.2 Combination of likelihoods of orthogonal analyses

As noted above, SMODELS now also provides the possibility to combine likelihoods from different analyses under the assumption that they are approximately uncorrelated. By approximately uncorrelated we mean that SRs do not overlap and inter-analyses correlations of systematic uncertainties (stemming from, e.g., luminosity measurements) can be neglected.[4] The

---

[4]Overlaps of SRs of one analysis with the control regions of another analysis in the combination can in principle induce correlations of systematic uncertainties and therefore should also be checked. However, we generally expect the effect to be negligible compared to other uncertainties in SMODELS.

combined likelihood $\mathcal{L}_C$ is then simply the product of the likelihoods $\mathcal{L}_i$ of the individual analyses. Furthermore, a common signal strength $\mu$ is assumed for all analyses. Thus

$$\mathcal{L}_C(\mu) = \prod_{i=1} \mathcal{L}_i(\mu s^i). \tag{6}$$

The individual likelihoods can correspond to best signal region likelihoods and/or combined signal region likelihoods from any of the three approaches explained above (turned on/off with the combineSRs=True/False switch). For the determination of the maximum likelihood, or more precisely the minimum negative log-likelihood $-\log\mathcal{L}_{\max} = -\log\mathcal{L}_C(\hat{\mu})$, scipy.optimize.minimize is used with the BFGS method. The resulting likelihood and $r$-values for the combination are displayed in the SMODELS output along with the individual results for each analysis.

As of now, the information of which analyses should be combined has to be provided by the user. This can be done in the parameters.ini file via the option combineAnas, providing a comma-separated list of two or more statistically independent analyses (identified by their analysis ID). A concrete example is

```
combineAnas = ATLAS-SUSY-2018-41,CMS-SUS-21-002
```

which will combine the two hadronic EW-ino searches from ATLAS and CMS. Generally, results from different experiments (ATLAS and CMS), different LHC runs (8 and 13 TeV), as well as fully hadronic and fully leptonic analyses can be regarded as approximately uncorrelated [16], at least within the approximations of SMODELS. For more sophisticated combinations, deeper scrutiny of the signal (and control) region definitions is needed; see [17] for an approach based on Monte Carlo simulation.

The combination of analyses is interesting for two reasons. First, the signal of a particular BSM scenario may be manifest in different final states, which are constrained by different analyses. Combining them uses more of the available data and thus provides more robust, and usually stronger, constraints. Second, experimental analyses can always be subject to over- or under-fluctuations of the backgrounds. In the former case, the observed limit is weaker, in the latter case stronger, than the expected limit. Again, the combination of different, approximately independent analyses can mitigate this effect and provide more robust constraints.

This is illustrated in Fig. 1 for a sample point from the EW-ino scan used in section 4, which features a bino-like $\tilde{\chi}_1^0$ with a mass of 257 GeV and wino-like $\tilde{\chi}_1^\pm$ and $\tilde{\chi}_2^0$ with masses of 617 GeV. The $\tilde{\chi}_1^\pm$ decays to 100% into $\tilde{\chi}_1^0 W^\pm$, while the $\tilde{\chi}_2^0$ decays to 96% into $\tilde{\chi}_1^0 h$ and to 4% into $\tilde{\chi}_1^0 Z$. The strongest constraints for this scenario come from the fully hadronic EW-ino searches using boosted $W$, $Z$ and Higgs bosons, ATLAS-SUSY-2018-41 [31] and CMS-SUS-21-002 [32]. Plotted in Fig. 1 are the likelihoods as a function of the signal strength $\mu$ for these two analyses and their combination, on the left for the expected and on the right for the observed data. As can be seen from the left panel, the CMS-SUS-21-002 analysis has the highest sensitivity and is expected to exclude the point with $r_{\exp} = 1.16$ if there is no new physics in the data (recall that $r \equiv 1/\mu_{95}$). The ATLAS analysis has slightly less sensitivity and is not expected to individually exclude the point ($\mu_{95} > 1$ or equivalently $r_{\exp} < 1$). Combining the likelihoods from both analyses (dashed blue curve in the plot), we arrive at an expected exclusion of $r_{\exp}(\text{combined}) = 1.52$, which illustrates the gain in sensitivity.

With the observed data, however, shown in the plot on the right, the strongest constraint comes from the ATLAS-SUSY-2018-41 analysis, excluding the point with an $r_{\text{obs}} = 1.32$. The CMS-SUS-21-002 analysis only gives an $r_{\text{obs}}$ value of 0.84. The reason is that the former analysis observed a deficit of events, while the latter observed a small excess. In an analysis-by-analysis approach, one could exclude the point based on the highest observed $r$-value, or conclude that it is still allowed because the most sensitive analysis (the one with highest

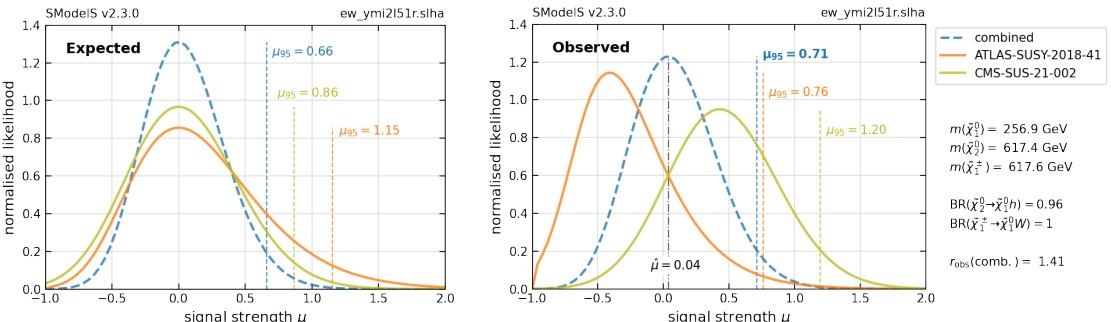

Figure 1: Visualisation of likelihoods for an EW-ino sample point with bino-like $\tilde{\chi}_1^0$ and wino-like $\tilde{\chi}_1^\pm$ and $\tilde{\chi}_2^0$ with masses $m_{\tilde{\chi}_1^0} = 257$ GeV, $m_{\tilde{\chi}_1^\pm, \tilde{\chi}_2^0} = 617$ GeV. Shown are the expected likelihoods (left panel) and observed likelihoods (right panel) as a function of the signal strength $\mu$ for the two hadronic EW-ino searches from ATLAS and CMS and their combination.

$r_{\mathrm{exp}}$) does not allow to exclude it. With the combined likelihood, however, one arrives at $r_{\mathrm{obs}}(\text{combined}) = 1.41$, thus robustly excluding the point.

We like to stress here that the expected limit always becomes stronger when combining. For the observed limit, however, the effect can go in either direction. The observed limit can become stronger, like in the example in Fig. 1. In the same way, it is possible that a point is excluded only by the combination of analyses, but not by any of the individual ones. But, the opposite can also happen, namely that a point excluded by some analysis becomes "unexcluded" by the combination with one or more other analyses. Indeed, all these cases will occur in the physics application in section 4.

An example code for visualising likelihoods and their combinations as in Fig. 1 is given in the How To's section of the online manual. Moreover, there are code examples showing how to compute the confidence level of an exclusion, and how to define and use a "combinability matrix" that describes which analyses and/or SRs are approximately orthogonal and thus combinable.

## 3 Database update

The database update with respect to v2.1.0 [7] concerns the results from 9 ATLAS and 12 CMS publications. Concretely, the following results were added corresponding to the availability of appropriate public numerical material on HEPDATA:[5]

**Gluino/squark searches:** In this category, we added UL results from two CMS analyses, the search in final states with jets plus leptons CMS-SUS-19-008 [33] and the search in final states with jets plus highly boosted $Z$-bosons CMS-SUS-19-013 [34]. From ATLAS, we added the gluino/squark UL and EM results from the search in final states with jets plus two leptons, ATLAS-SUSY-2018-05 [35].

**Stop/sbottom searches:** Here we added UL and EM results from two ATLAS publications: the stop search in the $2\ell$+jets channel, ATLAS-SUSY-2018-08 [36], and the sbottom search in final states with $b$-jets and taus (from $h \to \tau\tau$), ATLAS-SUSY-2018-40 [37]. On the CMS side,

---

[5]Unless specified otherwise, all results are from Run 2 of the LHC at $\sqrt{s} = 13$ TeV.

we added UL results from five publications, namely the $0\ell$ and $2\ell$ stop searches CMS-SUS-19-010 [38] and CMS-SUS-19-011 [39], the stop results from the search with soft leptons, CMS-SUS-18-004 [40], the sbottom results from CMS-SUS-18-007 [41], which uses $h \rightarrow \gamma\gamma$ decays, as well as the stop combination CMS-SUS-20-002 [42]. All these are results for full Run 2 luminosity: 139/fb for ATLAS and 137/fb for CMS. Moreover, lacking other EM results from CMS, we added the EMs which became available for CMS-SUS-16-050 [43], i.e. the $0\ell$ stop search for 35.9/fb.

**Electroweak searches:** These constitute the most important part of this database update. We added UL and EM results from ATLAS covering a large variety of EW-ino searches at Run 2; these are four searches in leptonic channels,[6] ATLAS-SUSY-2018-05 (2 leptons) [35], ATLAS-SUSY-2018-32 (2 OS leptons) [29], ATLAS-SUSY-2019-02 (soft leptons) [44] and ATLAS-SUSY-2019-09 (3 leptons) [30], as well as the fully hadronic search ATLAS-SUSY-2018-41 [31]. For all these, the combination of SRs is enabled either through a HISTFACTORY statistical model or a covariance matrix. It has to be stressed here that the HISTFACTORY models provided by ATLAS are a real boon and enormously benefit reinterpretation studies![7] For the ATLAS-SUSY-2018-41 analysis, no HISTFACTORY model is available so far, but ATLAS confirmed that the three SRs, for which EMs are available, are statistically independent, so we can trivially combine them assuming a diagonal covariance matrix. Last but not least, for completeness, we also added EM results for the search in final states with 3 leptons at 8 TeV, ATLAS-SUSY-2013-12 [45], which were previously missing and help cover low EW-ino masses. One analysis, which is not included although it does have extensive HEPDATA material, is the Run 2 search for electroweak production with compressed mass spectra ATLAS-SUSY-2018-16 [46]: the UL and EM results of this analysis depend on the assumed scenario, which conflicts with the default SMODELS assumptions.

On the CMS side, the most important additions are the UL and EM results from the fully hadronic EW-ino search, CMS-SUS-21-002 [32], and the search for higgsinos decaying to Higgs bosons (with $h \rightarrow b\bar{b}$), CMS-SUS-20-004 [23]. Both analyses provide correlation and covariance matrices allowing for SRs combination. While the purely Gaussian SLv1 [9] works well for CMS-SUS-21-002, the CMS-SUS-20-004 analysis has highly asymmetric background uncertainties in several SRs, which makes the SLv1 a bad approximation of the full likelihood [22]. Thankfully, the background uncertainties are reported in non-symmetrised form in the paper [23], and we can use this information to compute the skew terms for the SLv2; CMS-SUS-20-004 is thus the first analysis in the SMODELS database which uses the SLv2 following [18].

In addition to the above, we have implemented EW-ino UL results from CMS-SUS-18-007 [41], which is a search using Higgs boson to diphoton decays, and CMS-SUS-20-001 [47], a search for charginos and sleptons in final states with 2 leptons. It is deplorable, that other very relevant CMS analyses, like the EW-ino search in multi-lepton final states, CMS-SUS-19-012 (dedicated to Luc Pape!), and the search in $1\ell + h(\rightarrow b\bar{b})$ final states, CMS-SUS-20-003, provide no public material for reinterpretation or reuse.

**Long-lived particles:** The previous database already contained a large number of results from searches for long-lived particles, as these had been the primary target of the v2.1.0 update [7]. In v2.3.0, one new analysis is added to this: the 139/fb ATLAS search for heavy, long-lived, charged particles with $dE/dx$ measurement, ATLAS-SUSY-2018-42 [48]. Concretely, we implemented the UL results for HSCPs (long-lived staus) and R-hadrons (long-lived gluinos)

---

[6]The search in the $1\ell + \text{Higgs}(\rightarrow b\bar{b})$ channel, ATLAS-SUSY-2019-08 [28], was implemented in v1.2.4 [12].

[7]The signal patchsets typically provided together with the background-only statistical models are also a very good way of communicating acceptance×efficiency values for *all* signal and control regions without the need of an excessive number of auxiliary plots or tables.

together with R-hadron EMs for the two inclusive SRs; the implementation of an EM binning in target mass is left for a future release. The analysis has also considered simplified models for long-lived charginos, but, unfortunately, the material provided for this is not sufficient for being used by SMODELS [49].

**Recast with MadAnalysis5:** In addition to the UL and EM results listed above, which were directly provided by ATLAS and CMS, we generated a number of "home-grown" EMs through recasting with MADANALYSIS 5 [50, 51]. This helps fill gaps in EM coverage whenever a MAD-ANALYSIS 5 recast code is available. Concretely, we produced EMs for the 35.9/fb leptonic EW-ino searches CMS-SUS-16-039 [52, 53] and CMS-SUS-16-048 [54, 55], and the 137/fb multi-jet gluino and squark search CMS-SUS-19-006 [56, 57]. All three analyses come with a covariance matrix for SR combination in the SLv1 approach. The quality of the recasts is documented in [13].

With these additions, the SMODELS v2.3.0 database now comprises results from 38 ATLAS and 40 CMS searches at Run 2 ($\sqrt{s} = 13$ TeV), as well as 15 ATLAS and 18 CMS searches at Run 1 ($\sqrt{s} = 8$ TeV), covering a total of 111 experimental publications. 17 of the AT-LAS and 13 of the CMS searches are for full Run 2 luminosity. An overview of the complete database is given in Tables 1–3 in Appendix A and online at https://smodels.github.io/docs/ListOfAnalyses230. Validation plots for all results in the database are available online at https://smodels.github.io/docs/Validation230.

# 4 Physics impact

Since the largest part of the database update concerns searches for charginos and neutralinos, we demonstrate the physics impact of the new database and the new features in SMODELS by means of constraints on the EW-ino sector of the MSSM. To this end, we reuse the EW-ino scan points from [7]. In section 4.2 of that paper, the relevant Lagrangian parameters, i.e. the bino and wino mass parameters $M_1$ and $M_2$, the higgsino mass parameter $\mu$, and $\tan\beta = v_2/v_1$ were randomly scanned over within the following ranges:

$$
\begin{aligned}
10\,\text{GeV} < \quad & M_1 \quad < 3\,\text{TeV}\,, \\
100\,\text{GeV} < \quad & M_2 \quad < 3\,\text{TeV}\,, \\
100\,\text{GeV} < \quad & \mu \quad < 3\,\text{TeV}\,, \\
5 < \quad & \tan\beta \quad < 50\,.
\end{aligned}
\tag{7}
$$

The other SUSY breaking parameters were fixed to 10 TeV.[8] The lower limits on $M_2$ and $\mu$ were chosen so as to avoid the LEP constraints on light charginos, and the lightest neutralino $\tilde{\chi}_1^0$ was required to be the lightest supersymmetric particle (LSP). The mass spectra and decay tables were computed with SOFTSUSY 4.1.11 [58, 59], which includes the $\tilde{\chi}_1^\pm \to \pi^\pm \tilde{\chi}_1^0$ decay calculation following [60] for small mass differences below about 1.5 GeV. Cross sections were computed first at leading order (LO) with PYTHIA 8 [61, 62], and reevaluated at next-to-LO with PROSPINO [63] for all points which had $r_{\text{max}} \equiv \max(r_{\text{obs}}) > 0.7$ in SMODELS v2.1.0 with the LO cross sections.

From close to 100k points of the complete scan in [7], we here select the subset of points with only prompt decays (no long-lived particles, all decay widths $\Gamma_{\text{tot}} > 10^{-11}$ GeV). Moreover, we require $m_{\tilde{\chi}_1^0} < 500$ GeV and $m_{\tilde{\chi}_1^\pm} < 1200$ GeV in order to focus on the region which the

---

[8]We assume that parameters can always be adjusted in the stop sector such that $m_h \simeq 125$ GeV without influencing the EW-ino sector.

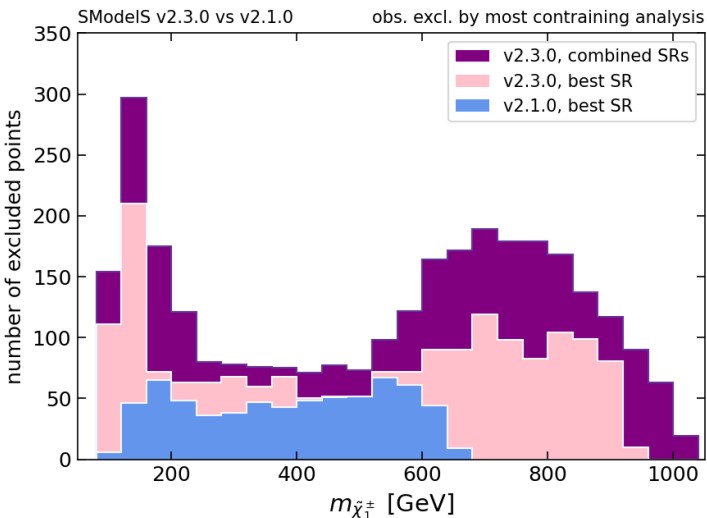

Figure 2: Comparison of exclusion power of SMODELS v2.3.0 versus v2.1.0 as a function of the lighter chargino mass, $m_{\tilde{\chi}_1^\pm}$. Shown are the number of points excluded by the most constraining analysis, i.e. the number of points with $r_{\max} \equiv \max(r_{\text{obs}}) \geq 1$.

current prompt EW-ino searches are sensitive to. This leaves us with 18544 points, which we analyse with SMODELS v2.3.0.

The increase of exclusion power as compared to [7] is illustrated in Figure 2, which shows the number of excluded points as a function of the $\tilde{\chi}_1^\pm$ mass. The plot compares v2.1.0 to v2.3.0 with and without SR combination.[9,10] Different analyses are not combined at this stage. We note that the new experimental results in v2.3.0 extend the reach in chargino mass by about 300 GeV in the "best SR" approach (i.e., when SRs are not combined). Combination of SRs extends this reach by another 100 GeV, but mostly it increases the number of excluded points in the $m_{\tilde{\chi}_1^\pm} \approx 500$–1000 GeV mass range; this concerns primarily scenarios with a bino-like LSP. The effect comes mostly from the ATLAS-SUSY-2018-41 analysis, where the combination of SRs allows to simultaneously take into account the signal contributions to $VV$, $Vh$ and $hh$ final states ($V = W^\pm, Z$), thus increasing the constraining power of the search. Finally, there is also a significant increase in the number of excluded points at low $m_{\tilde{\chi}_1^\pm} \lesssim 200$ GeV; these are to large extent points with a higgsino-like LSP. Overall, the number of excluded points increases from 661 with v2.1.0 to 2974 (1787) with v2.3.0 when SR combination is turned on (off).

Given Figure 2, it is interesting to ask which experimental results are driving the exclusion in different regions of the parameter space. To answer this question, Figure 3 shows the points excluded by the LHC searches in the SMODELS v2.3.0 database in the $m_{\tilde{\chi}_1^\pm}$ versus $m_{\tilde{\chi}_1^0}$ plane. The color of each excluded point denotes the most constraining analysis, that is the analysis giving the highest observed $r$-value. As can be seen, in the low mass range, $m_{\tilde{\chi}_1^\pm} \lesssim 500$ GeV, the constraints come from a variety of different analyses, while the high mass range, $m_{\tilde{\chi}_1^\pm} \gtrsim 500$ GeV is completely dominated by the ATLAS-SUSY-2018-41 analysis. The reason for this is that the ATLAS-SUSY-2018-41 analysis observed less events than expected (at least in the three super signal regions for which EMs are available) and therefore sets stronger limits than expected, and also stronger limits than the equivalent analysis from CMS, CMS-SUS-21-002, which saw a small excess in events.

This brings us to the issue that the most constraining analysis is not necessarily also the most sensitive one. From the statistics point of view, however, as long as limits are set on an

---

[9]The combination of SRs is turned on/off by setting `combineSRs=True/False` in the `parameters.ini` file.
[10]All our results were obtained with `sigmacut=1e-3 fb`.

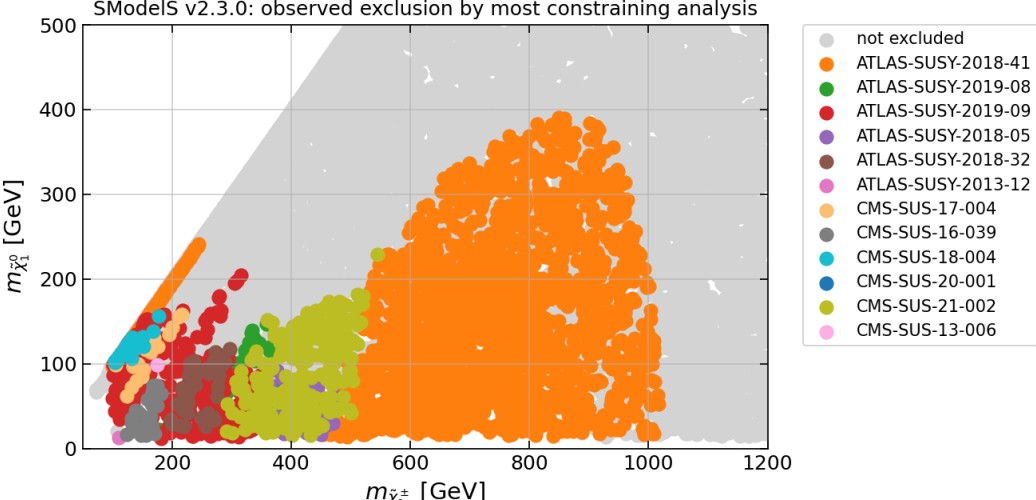

Figure 3: Scan points excluded by the most constraining analysis in SMODELS v2.3.0, with combination of SRs turned on. The colour denotes the analysis that gives the highest $r$-value (see legend). Grey points are not excluded.

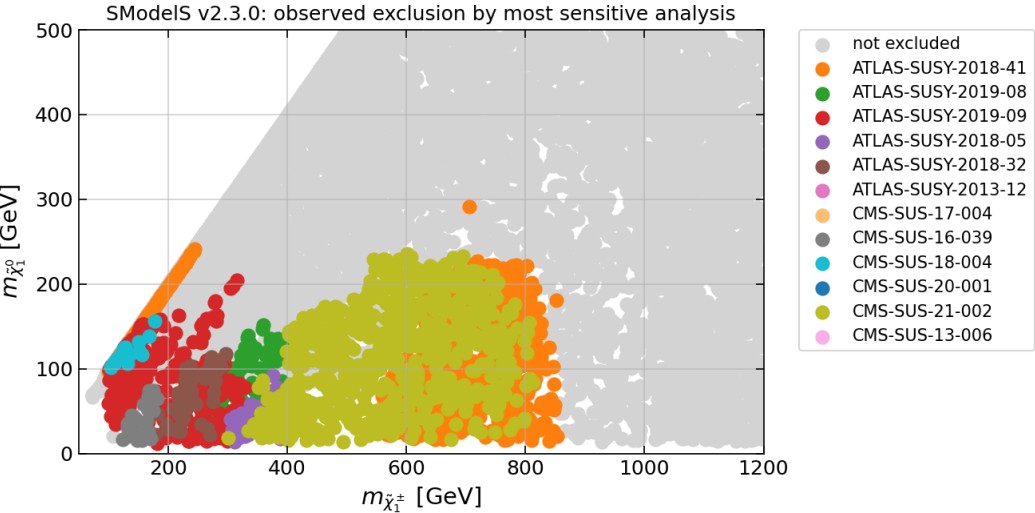

Figure 4: As Fig. 3 but for points excluded by the most sensitive analysis.

analysis-by-analysis basis, using only the most sensitive result is the more rigorous approach in order to stay at the 95% confidence level. Indeed, as can be seen in Fig. 4, the picture changes quite significantly when considering the exclusion from the most sensitive analysis only. In particular when the most sensitive analysis is either the ATLAS or the CMS search in fully hadronic final states, the excluded region shrinks from $(m_{\tilde{\chi}_1^\pm}, m_{\tilde{\chi}_1^0}) \lesssim (1000, 400)$ GeV in Fig. 3 to $(m_{\tilde{\chi}_1^\pm}, m_{\tilde{\chi}_1^0}) \lesssim (850, 250)$ GeV in Fig. 4.

The assessment of the excluded parameter space can be improved by statistically combining the relevant analyses, as already demonstrated in Fig. 1 for a specific benchmark point. The results in Figs. 3 and 4 indeed motivate a combination of the two hadronic EW-ino searches, ATLAS-SUSY-2018-41 and CMS-SUS-21-002, as they are the most sensitive and/or most constraining ones in the high $m_{\tilde{\chi}_1^\pm}$ range (roughly for $m_{\tilde{\chi}_1^\pm} \gtrsim 400$ GeV). Figure 5 shows how the combination of these two analyses improves the sensitivity to EW-ino signals as compared to

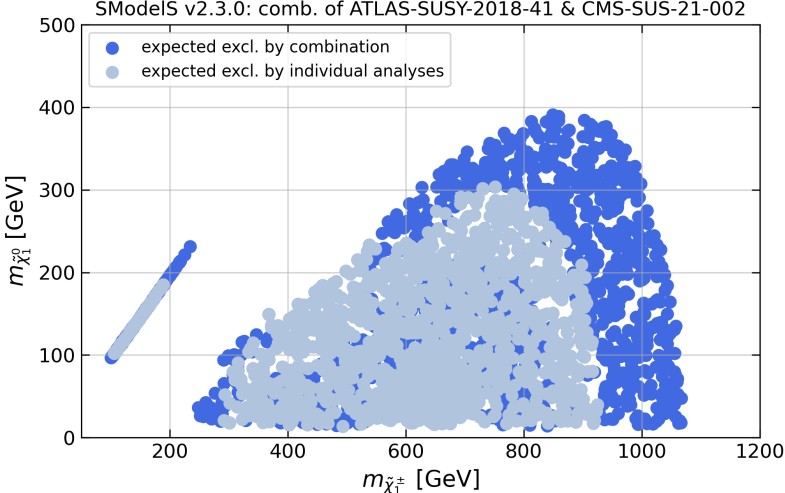

Figure 5: Effect of statistically combining the hadronic EW-ino searches from ATLAS and CMS, ATLAS-SUSY-2018-41 and CMS-SUS-21-002 on the **expected reach**: in light blue the expected exclusion of the individual ATLAS or CMS analyses, in dark blue the expected exclusion of the combination.

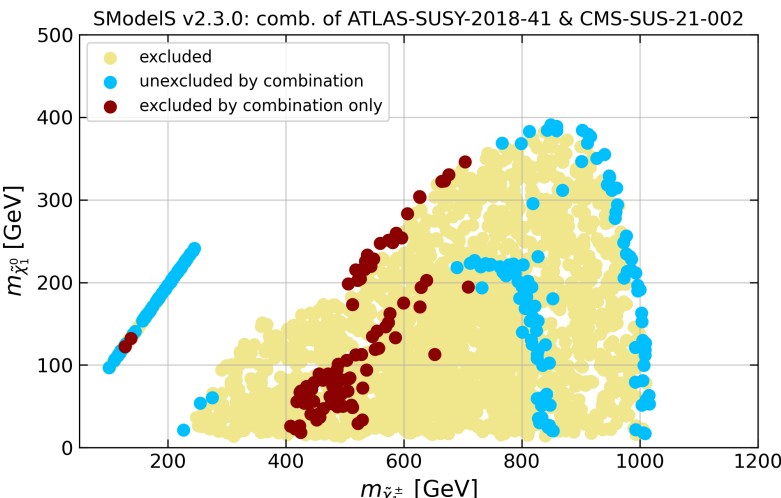

Figure 6: Effect of statistically combining the hadronic EW-ino searches from ATLAS and CMS, ATLAS-SUSY-2018-41 and CMS-SUS-21-002 on the **observed limit**. The khaki (light yellowish) and dark red coloured points are excluded at 95% confidence level by the combination, with the dark red points being those which are excluded only by the combination but not by one of the two analyses alone. The light blue points would be excluded by either ATLAS-SUSY-2018-41 or CMS-SUS-21-002, but are not excluded any more in the combination.

the single analysis approach: the expected limits are extended by about 200 GeV in chargino mass and by up to 100 GeV in LSP mass. The effect on the observed exclusion is shown Figure 6. Here, the khaki (light yellowish) and dark red coloured points are excluded by the combination, with the dark red points being those which are excluded only by the combination but not by either the ATLAS or the CMS analysis alone. In contrast, the light blue points would be excluded by one analysis, but are not excluded any more in the combination.

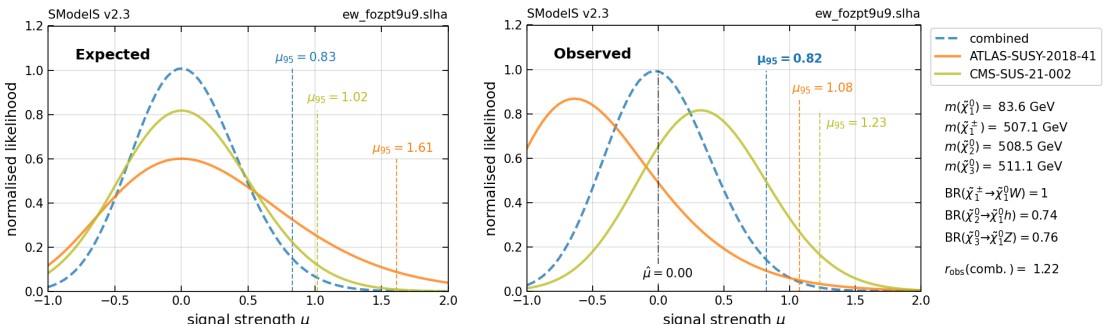

Figure 7: Visualisation of likelihoods for one of the dark red points from Fig. 6, i.e. a point which is only excluded by the combination of the considered ATLAS and CMS analyses, but not by any of the two analyses alone. See text for details.

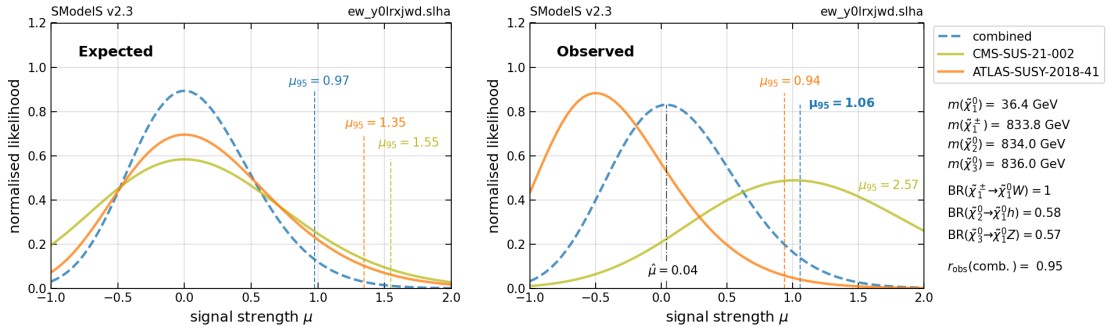

Figure 8: Visualisation of likelihoods for one of the light blue points from Fig. 6, concretely a point which is excluded by the ATLAS analysis, but not any more so when combining it with the CMS analysis. See text for details.

Concrete examples for a dark red and a light blue point from Fig. 6 are given in Figs. 7 and 8, respectively. The sample point in Fig. 7 lies at $(m_{\tilde{\chi}_1^\pm}, m_{\tilde{\chi}_1^0}) = (507, 84)$ GeV, the point in Fig. 8 at $(m_{\tilde{\chi}_1^\pm}, m_{\tilde{\chi}_1^0}) = (834, 36)$ GeV. Both feature a bino-like $\tilde{\chi}_1^0$ and higgsino-like $\tilde{\chi}_1^\pm, \tilde{\chi}_{2,3}^0$. Moreover, in both cases neither the ATLAS nor the CMS hadronic analysis is expected to exclude the point; only the combination of both analyses gives high enough sensitivity. For the point in Fig. 7, this behaviour is replicated also with the observed data: indeed we find $r_{\text{obs}} = 0.81$ and $0.93$ for the CMS and ATLAS analysis, respectively, so neither analysis individually excludes the point (recall that $r = 1/\mu_{95}$). In the combination, this moves to $r_{\text{obs}}(\text{comb.}) = 1.22$, resulting in a solid exclusion. In contrast, in the example in Fig. 8, the behaviour of the observed limits is quite different: the ATLAS analysis excludes the point with an $r_{\text{obs}}$ of 1.07, while the CMS analysis, although having very similar sensitivity, gives only $r_{\text{obs}} = 0.39$. Combining the two analyses results in $r_{\text{obs}}(\text{comb.}) = 0.95$, which means the point is not excluded any more.

Altogether, the combination of the ATLAS and CMS hadronic EW-ino searches excludes 2258 of the scan points. Of the remaining points, further 614 are excluded by some other EW-ino search. This gives a total of 2872 excluded points, as compared to 2974 (2048) when considering the limit from the most constraining (most sensitive) analysis only. Indeed the overall change in the observed limit when comparing Fig. 6 to Fig. 3 is rather small. This is due to the data under-fluctuations (with respect to the expected background) seen by the hadronic ATLAS search and the over-fluctuations seen by the CMS one. The combined limit is, however, statistically more reliable, since it makes use of a larger amount of data coming from both the ATLAS and CMS searches.

A comment is in order regarding the bino, wino and higgsino mixing. The bulk of the excluded points at $m_{\tilde{\chi}_1^\pm} \approx 200–1000$ GeV in Fig. 6 has a (mostly) bino-like LSP, with the next heavier states being mostly higgsino- or wino-like; highly mixed charginos and neutralinos make only a fraction of the scan points, due to volume effects. As a consequence, the dark red and light blue points form two "arcs" in the $m_{\tilde{\chi}_1^\pm}$ versus $m_{\tilde{\chi}_1^0}$ plane, the lower one featuring higgsino-like $\tilde{\chi}_1^\pm$ and $\tilde{\chi}_{2,3}^0$ and the higher one featuring wino-like $\tilde{\chi}_1^\pm$ and $\tilde{\chi}_2^0$. The few dark red/light blue points scattered away from these arcs have significant wino-higgsino mixing. The isolated diagonal line of points at $m_{\tilde{\chi}_1^\pm} \approx 100–250$ GeV, on the other hand, is characterised by a higgsino-like LSP, with the signal coming from the production of wino-like $\tilde{\chi}_2^\pm$ and $\tilde{\chi}_3^0$. In Appendix D we provide auxiliary results in terms of the relevant MSSM parameters, which illustrate the above statements.

All in all, the combination of analyses, here illustrated by the example of combining the fully hadronic EW-ino searches from ATLAS and CMS, leads to better and statistically more robust constraints: first, combinations of analyses use more of the available data thus increasing the sensitivity to the BSM signal; second, while individual analyses can lead to an over(under)-aggressive exclusion of the parameter space due to under(over)-fluctuations of the data with respect to the expected background, the combination of analyses reduces the impact of such fluctuations.

# 5 Conclusions

The wealth of experimental constraints on new physics from Run 2 of the LHC makes it increasingly important to be able to perform global (re)interpretation studies. The aim is to find out on the one hand how all the available experimental information constrains complicated scenarios, and on the other hand what are the most likely regions where new physics can still manifest itself at the LHC. This requires software tools that encompass searches in as many final states as possible, and that can build global likelihoods from them.

The SMODELS package is a public tool that can be useful in this endeavour. It is designed for the fast reinterpretation of LHC searches for new physics on the basis of simplified-model results, mostly stemming from searches for supersymmetric particles. Owing to its speed, it is particularly well suited for large scans and model surveys.

Version 2.3 of SMODELS, presented in this paper, comes with a significant database update with the latest available experimental results for full Run 2 luminosity. In total, the SMODELS database now comprises results from 38 ATLAS and 40 CMS searches at Run 2, as well as 15 ATLAS and 18 CMS searches at Run 1, covering a total of 111 experimental publications. 17 of the ATLAS and 13 of the CMS searches are for full Run 2 luminosity. This comprises in particular the full suit of available and reusable electroweak-ino searches from Run 2.

On the code side, the most important new feature is the ability to combine likelihoods from different analyses. This enables statistically more rigorous constraints and opens the way for global likelihood analyses for LHC searches. We reviewed the different ways of computing likelihoods for a signal hypothesis from the publicly available information, and explained how likelihoods can be combined in SMODELS.

The physics impact was demonstrated by means of a case study of the electroweak-ino sector of the MSSM. With the new database, the reach in chargino mass increases from about 650 GeV in [7] to about 1 TeV in v2.3.0. for promptly decaying EW-inos. The total number of excluded points increases by more than a factor of four with respect to [7]. Last but not least, we exemplified how the combination of approximately independent analyses helps average out statistical fluctuations and thus leads to more robust limits. An extensive global study of the EW-ino sector of the MSSM is in preparation.

**Data management:** SMODELS is public software distributed under the GNU General Public License v3 (GPLv3). It is available on GitHub and in the Python Package Index (PyPI). The SMODELS database is available in text form on GitHub and as a binary pickle file on Zenodo [64]. The complete dataset (input and output files) from the EW-ino scan used in this paper is also made available on Zenodo [65], ensuring full reproducibility of the results presented in this paper.

# Acknowledgments

We thank Jack Araz for identifying an issue with the expected upper limits in the simplified likelihoods computation and for help with cleaning up the statistics code. Moreover, we thank Shion Chen for very useful exchanges on ATLAS-SUSY-2018-41, and William Ford for in-depth discussions on CMS-SUS-20-004 and the SLv2, and for assuring that additional auxiliary material was made available on HEPDATA for this analysis.

We also gratefully acknowledge numerous exchanges with the ATLAS and CMS SUSY group conveners: their engagement is remarkable, and we hope they will not tire responding to our questions and/or forwarding them to the relevant people. Last but not least, we warmly thank all ATLAS and CMS analysers who ensure that their work can be reused by others—without their dedication to make extensive auxiliary material available on HEPDATA, public tools like SMODELS, and reinterpretation studies in general, would not be possible.

**Funding information** This work was supported in part by the IN2P3 master project "Théorie – BSMGA", by the French Agence Nationale de la Recherche (ANR) under grant ANR-21-CE31-0023 (PRCI SLDNP) and the Austrian Science Fund (FWF) under grant number I 5767-N. T.P. is supported by the Initiatives de Recherche à Grenoble Alpes (IRGA) ANR-15-IDEX-02 project no. G7H-IRG21B26 (APM@LHC). A.L. is supported by FAPESP grant no. 2018/25225-9 and 2021/01089-1.

# A  Complete list of experimental results in the v2.3.0 database

Table 1: List of ATLAS Run 2 analyses and their types of results in the SMODELS v2.3.0 database. Apart from the HSCP, disappearing tracks and displaced lepton searches, all analyses require $\not{E}_T$ in the final state (for conciseness omitted in the short descriptions). EWK stands for electroweak (-ino or slepton) production. The last column specifies whether and how SRs are combined. New additions are highlighted in bold.

| ID | Short Description | $\mathcal{L}$ [fb$^{-1}$] | UL$_{\text{obs}}$ | UL$_{\text{exp}}$ | EM | comb. |
|---|---|---|---|---|---|---|
| ATLAS-SUSY-2015-01 [66] | 2 $b$-jets | 3.2 | ✓ | | | |
| ATLAS-SUSY-2015-02 [67] | 1$\ell$ stop | 3.2 | ✓ | | ✓ | |
| ATLAS-SUSY-2015-06 [68] | 0$\ell$ + 2–6 jets | 3.2 | | | ✓ | |
| ATLAS-SUSY-2015-09 [69] | jets + 2 SS or $\geq 3\ell$ | 3.2 | ✓ | | | |
| ATLAS-SUSY-2016-06 [70] | disappearing tracks | 36.1 | | | ✓ | |
| ATLAS-SUSY-2016-07 [71] | 0$\ell$ + jets | 36.1 | ✓ | | ✓ | |
| ATLAS-SUSY-2016-08 [72] | displaced vertices | 32.8 | ✓ | | | |
| ATLAS-SUSY-2016-14 [73] | 2 SS or 3 $\ell$'s + jets | 36.1 | ✓ | | | |
| ATLAS-SUSY-2016-15 [74] | 0$\ell$ stop | 36.1 | ✓ | | | |
| ATLAS-SUSY-2016-16 [75] | 1$\ell$ stop | 36.1 | ✓ | | ✓ | |
| ATLAS-SUSY-2016-17 [76] | 2 OS $\ell$ | 36.1 | ✓ | | | |
| ATLAS-SUSY-2016-19 [77] | 2 $b$-jets + $\tau$'s | 36.1 | ✓ | | | |
| ATLAS-SUSY-2016-24 [78] | 2–3 $\ell$'s, EWK | 36.1 | ✓ | | ✓ | |
| ATLAS-SUSY-2016-26 [79] | $\geq 2c$-jets | 36.1 | ✓ | | | |
| ATLAS-SUSY-2016-27 [80] | jets + $\gamma$ | 36.1 | ✓ | | ✓ | |
| ATLAS-SUSY-2016-28 [81] | 2 $b$-jets | 36.1 | ✓ | | | |
| ATLAS-SUSY-2016-32 [82] | HSCP | 31.6 | ✓ | ✓ | ✓ | |
| ATLAS-SUSY-2016-33 [83] | 2 SFOS $\ell$'s | 36.1 | ✓ | | | |
| ATLAS-SUSY-2017-01 [84] | $Wh(bb)$, EWK | 36.1 | ✓ | | | |
| ATLAS-SUSY-2017-02 [85] | 0$\ell$ + jets | 36.1 | ✓ | ✓ | | |
| ATLAS-SUSY-2017-03 [86] | multi-$\ell$ EWK | 36.1 | ✓ | | ✓ | |
| ATLAS-SUSY-2018-04 [87] | 2 hadronic taus | 139.0 | ✓ | | ✓ | PYHF |
| **ATLAS-SUSY-2018-05 [35]** | 2$\ell$ + jets, EWK | 139.0 | ✓ | | ✓ | PYHF |
| **ATLAS-SUSY-2018-05 [35]** | 2$\ell$ + jets, strong | 139.0 | | | ✓ | |
| ATLAS-SUSY-2018-06 [88] | 3$\ell$, EWK | 139.0 | ✓ | ✓ | ✓ | |
| **ATLAS-SUSY-2018-08 [36]** | 2 OS $\ell$ | 139.0 | ✓ | | ✓ | |
| ATLAS-SUSY-2018-10 [89] | 1$\ell$ + jets | 139.0 | ✓ | | ✓ | |
| ATLAS-SUSY-2018-12 [90] | 0$\ell$ + jets | 139.0 | ✓ | ✓ | ✓ | |
| ATLAS-SUSY-2018-14 [91] | displaced vertices | 139.0 | | | ✓ | PYHF |
| ATLAS-SUSY-2018-22 [92] | multi-jets | 139.0 | ✓ | | ✓ | |
| ATLAS-SUSY-2018-23 [93] | $Wh(\gamma\gamma)$, EWK | 139.0 | ✓ | ✓ | | |
| ATLAS-SUSY-2018-31 [94] | $2b + 2h(bb)$ | 139.0 | ✓ | | ✓ | PYHF |
| **ATLAS-SUSY-2018-32 [29]** | 2 OS $\ell$ | 139.0 | ✓ | | ✓ | PYHF |
| **ATLAS-SUSY-2018-40 [37]** | $2b + 2h(\tau\tau)$ | 139.0 | ✓ | ✓ | ✓ | |
| **ATLAS-SUSY-2018-41 [31]** | hadr. EWK search | 139.0 | ✓ | ✓ | ✓ | SLv1 |
| **ATLAS-SUSY-2018-42 [48]** | charged LLPs, dE/dx | 139.0 | ✓ | ✓ | ✓ | |
| **ATLAS-SUSY-2019-02 [44]** | 2 soft $\ell$'s, EWK | 139.0 | ✓ | | ✓ | SLv1 |
| ATLAS-SUSY-2019-08 [28] | 1$\ell$ + $h(bb)$, EWK | 139.0 | ✓ | | ✓ | PYHF |
| **ATLAS-SUSY-2019-09 [30]** | 3$\ell$, EWK | 139.0 | ✓ | ✓ | ✓ | PYHF |

Table 2: List of CMS Run 2 analyses and their types of results in the SMODELS v2.3.0 database. Apart from the HSCP, disappearing tracks and displaced lepton searches, all analyses require $\not{E}_T$ in the final state (for conciseness omitted in the short descriptions). EWK stands for electroweak (-ino or slepton) production. The last column specifies whether and how SRs are combined. New additions are highlighted in bold.

| ID | Short Description | $\mathcal{L}$ [fb$^{-1}$] | UL$_{\text{obs}}$ | UL$_{\text{exp}}$ | EM | comb. |
|---|---|---|---|---|---|---|
| CMS-PAS-EXO-16-036 [95] | HSCP | 12.9 | ✓ | | | |
| CMS-PAS-SUS-16-052 [96] | ISR jet + soft $\ell$ | 35.9 | ✓ | | ✓ | SLv1 |
| CMS-SUS-16-009 [97] | $0\ell$ + jets, top tag | 2.3 | ✓ | ✓ | | |
| CMS-SUS-16-032 [98] | 2 $b$- or 2 $c$-jets | 35.9 | ✓ | | | |
| CMS-SUS-16-033 [99] | $0\ell$ + jets | 35.9 | ✓ | ✓ | ✓ | |
| CMS-SUS-16-034 [100] | 2 SFOS $\ell$ | 35.9 | ✓ | | | |
| CMS-SUS-16-035 [101] | 2 SS $\ell$ | 35.9 | ✓ | | | |
| CMS-SUS-16-036 [102] | $0\ell$ + jets | 35.9 | ✓ | ✓ | | |
| CMS-SUS-16-037 [103] | $1\ell$ + jets with MJ | 35.9 | ✓ | | | |
| CMS-SUS-16-039 [52] | multi-$\ell$, EWK | 35.9 | ✓ | | ✓ | SLv1 |
| CMS-SUS-16-041 [104] | multi-$\ell$ + jets | 35.9 | ✓ | | | |
| CMS-SUS-16-042 [105] | $1\ell$ + jets | 35.9 | ✓ | | | |
| CMS-SUS-16-043 [106] | $Wh(bb)$, EWK | 35.9 | ✓ | | | |
| CMS-SUS-16-045 [107] | $2\,b + 2\,h(\gamma\gamma)$ | 35.9 | ✓ | | | |
| CMS-SUS-16-046 [108] | high-$p_T$ $\gamma$ | 35.9 | ✓ | | | |
| CMS-SUS-16-047 [109] | $\gamma$ + jets, high $H_T$ | 35.9 | ✓ | | | |
| CMS-SUS-16-048 [54] | 2 OS $\ell$, soft | 35.9 | | | ✓ | SLv1 |
| **CMS-SUS-16-050 [43]** | $0\ell$ + top tag | 35.9 | ✓ | ✓ | ✓ | SLv1 |
| CMS-SUS-16-051 [110] | $1\ell$ stop | 35.9 | ✓ | ✓ | | |
| CMS-SUS-17-003 [111] | 2 taus | 35.9 | ✓ | | | |
| CMS-SUS-17-004 [112] | EWK combination | 35.9 | ✓ | | | |
| CMS-SUS-17-005 [113] | $1\ell$ + jets, top tag | 35.9 | ✓ | ✓ | | |
| CMS-SUS-17-006 [114] | jets + boosted $h(bb)$ | 35.9 | ✓ | ✓ | | |
| CMS-SUS-17-009 [115] | SFOS $\ell$ | 35.9 | ✓ | ✓ | | |
| CMS-SUS-17-010 [116] | $2\ell$ stop | 35.9 | ✓ | ✓ | | |
| CMS-SUS-18-002 [117] | $\gamma$ + ($b$-)jets, top tag | 35.9 | ✓ | ✓ | | |
| **CMS-SUS-18-004 [40]** | 2–3 soft $\ell$'s | 137.0 | ✓ | ✓ | | |
| **CMS-SUS-18-007 [41]** | $2h(\gamma\gamma)$, EWK | 77.5 | ✓ | ✓ | | |
| CMS-EXO-19-001 [118] | non-prompt jets | 137.0 | | | ✓ | |
| CMS-EXO-19-010 [119] | disappearing tracks | 101.0 | | | ✓ | |
| **CMS-SUS-19-006 [56]** | $0\ell$ + jets, MHT | 137.0 | ✓ | ✓ | ✓ | SLv1 |
| **CMS-SUS-19-008 [33]** | 2–3$\ell$ + jets | 137.0 | ✓ | ✓ | | |
| CMS-SUS-19-009 [120] | $1\ell$ + jets, MHT | 137.0 | ✓ | ✓ | | |
| **CMS-SUS-19-010 [38]** | jets + top and $W$-tag | 137.0 | ✓ | ✓ | | |
| **CMS-SUS-19-011 [39]** | $2\ell$ stop | 137.0 | ✓ | ✓ | | |
| **CMS-SUS-19-013 [34]** | jets + boosted $Z$'s | 137.0 | ✓ | ✓ | | |
| **CMS-SUS-20-001 [47]** | SFOS $\ell$ | 137.0 | ✓ | ✓ | | |
| **CMS-SUS-20-002 [42]** | stop combination | 137.0 | ✓ | ✓ | | |
| **CMS-SUS-20-004 [23]** | $2\,h(bb)$, EWK | 137.0 | ✓ | ✓ | ✓ | SLv2 |
| **CMS-SUS-21-002 [32]** | hadr. EWK search | 137.0 | ✓ | ✓ | ✓ | SLv1 |

Table 3: List of 15 ATLAS and 18 CMS Run 1 analyses and their types of results in the SMODELS v2.3.0 database. Apart from the HSCP searches, all analyses require $\not{E}_T$ in the final state (for conciseness omitted in the short descriptions). EWK stands for electroweak(-ino) production. New in v2.3.0 are the EMs for ATLAS-SUSY-2013-12. The column **comb.** from Tables 1–2 is omitted because none of the Run 1 analyses provides any information on background correlations.

| ID | Short Description | $\mathcal{L}$ [fb$^{-1}$] | UL$_{\text{obs}}$ | UL$_{\text{exp}}$ | EM |
|---|---|---|---|---|---|
| ATLAS-SUSY-2013-02 [121] | $0\ell$ + 2–6 jets | 20.3 | ✓ | | ✓ |
| ATLAS-SUSY-2013-04 [122] | $0\ell$ + 7–10 jets | 20.3 | ✓ | | ✓ |
| ATLAS-SUSY-2013-05 [123] | $0\ell$ + $2b$-jets | 20.1 | ✓ | | ✓ |
| ATLAS-SUSY-2013-08 [124] | $Z(\ell\ell)$ + $b$-jets | 20.3 | ✓ | | |
| ATLAS-SUSY-2013-09 [125] | 2 SS $\ell$ + 0–3 $b$-jets | 20.3 | ✓ | | ✓ |
| ATLAS-SUSY-2013-11 [126] | $2\ell$ $(e,\mu)$, EWK | 20.3 | ✓ | | ✓ |
| **ATLAS-SUSY-2013-12 [45]** | $3\ell$ $(e,\mu,\tau)$, EWK | 20.3 | ✓ | | ✓ |
| ATLAS-SUSY-2013-15 [127] | $1\ell$ + 4 $(1b)$ jets | 20.3 | ✓ | | ✓ |
| ATLAS-SUSY-2013-16 [128] | $0\ell$ + 6 $(2b)$ jets | 20.1 | ✓ | | ✓ |
| ATLAS-SUSY-2013-18 [129] | jets + $\geq 3b$-jets | 20.1 | ✓ | | ✓ |
| ATLAS-SUSY-2013-19 [130] | 2 OS $\ell$ + $(b$-)jets | 20.3 | ✓ | | |
| ATLAS-SUSY-2013-20 [131] | $2\ell$ $(e,\mu)$ + jets | 20.3 | ✓ | | |
| ATLAS-SUSY-2013-21 [132] | monojet or $c$-jet | 20.3 | | | ✓ |
| ATLAS-SUSY-2013-23 [133] | $1\ell$ + 2 $b$-jets (or $2\gamma$) | 20.3 | ✓ | | |
| ATLAS-SUSY-2014-03 [134] | $\geq 2$ $(c$-)jets | 20.3 | | | ✓ |
| CMS-EXO-12-026 [135] | HSCP | 18.8 | ✓ | | |
| CMS-EXO-13-006 [136] | HSCP | 18.8 | | | ✓ |
| CMS-PAS-SUS-13-015 [137] | $\geq 5$ $(1b)$ jets | 19.4 | | | ✓ |
| CMS-PAS-SUS-13-016 [138] | $2\ell$ + $\geq 4$ $(2b)$ jets | 19.7 | ✓ | | ✓ |
| CMS-PAS-SUS-13-018 [139] | 1–2 $b$-jets, $M_{\text{CT}}$ | 19.4 | ✓ | | |
| CMS-PAS-SUS-13-023 [140] | $0\ell$ stop | 18.9 | ✓ | | |
| CMS-SUS-12-024 [141] | $0\ell$ + $\geq 3$ $(1b)$ jets | 19.4 | ✓ | | ✓ |
| CMS-SUS-12-028 [142] | multi $(b$-)jets, $\alpha_T$ | 11.7 | ✓ | ✓ | |
| CMS-SUS-13-002 [143] | $\geq 3\ell$ (+jets) | 19.5 | ✓ | ✓ | |
| CMS-SUS-13-004 [144] | $\geq 1b$-jet, Razor | 19.3 | ✓ | | |
| CMS-SUS-13-006 [145] | multi-$\ell$, EWK | 19.5 | ✓ | | |
| CMS-SUS-13-007 [146] | $1\ell$ + $\geq 2b$-jets | 19.3 | ✓ | | ✓ |
| CMS-SUS-13-011 [147] | $1\ell$ + $\geq 4$ $(1b)$jets | 19.5 | ✓ | | ✓ |
| CMS-SUS-13-012 [148] | jets + $\not{H}_T$ | 19.5 | ✓ | ✓ | ✓ |
| CMS-SUS-13-013 [149] | 2 SS $\ell$ + $(b$-)jets | 19.5 | ✓ | ✓ | ✓ |
| CMS-SUS-13-019 [150] | $\geq 2$ jets, $M_{\text{T2}}$ | 19.5 | ✓ | | |
| CMS-SUS-14-010 [151] | $b$-jets + $4W$ | 19.5 | ✓ | ✓ | |
| CMS-SUS-14-021 [152] | soft $\ell$, low jet mult. | 19.7 | ✓ | ✓ | ✓ |

# B   Database add-ons

When running SMODELS, the user has to specify which database to use. This is done in the `parameters.ini` file by giving the path to the `smodels-database` folder, the path to a pickle file or a URL path. Details are explained in the Using SMODELS section of the online manual. The available databases can be seen on the smodels-database-release page on github.

Shorthand notations are available: `path=official` refers to the official database of the users' SMODELS version, while `path=latest` refers to the latest available database release. The '+' operator allows for extending the "official" or "latest" database with add-ons:

**+fastlim:** adds fastlim results (from early 8 TeV ATLAS analyses); from v2.1.0 onward

**+superseded:** adds results which were previously available but were superseded by newer ones; from v2.1.0 onward

**+nonaggregated:** adds analyses with non-aggregated SRs in addition to the aggregated results in CMS analyses; from v2.2.0 onward

**+full_llhds:** replaces simplified HISTFACTORY statistical models by full ones in ATLAS analyses; from v2.3.0 onward (careful, this increases a lot the runtime!)

Examples are "official+nonaggregated" or "official+nonaggregated+full_llhds". Note that order matters: results are replaced in the specified sequence, so "full_llhds+official" will fall back onto the official database with simplified HISTFACTORY statistical models. The add-ons can also be used alone, e.g. `path=full_llhds`, though this is of limited practical use. Finally, "debug" refers to a version of the database with extra information that is however not intended for usage by a regular user and only mentioned here for completeness.

# C   Examples combining more than two analyses

The benchmark point used in Fig. 1 is in fact constrained not only by the two hadronic EW-ino searches ATLAS-SUSY-2018-41 [31] and CMS-SUS-21-002 [32], but also by the search for $\tilde{\chi}_1^\pm \tilde{\chi}_2^0 \to Wh + \not{E}_T$ in the $1\ell + b\bar{b} + \not{E}_T$ final state, ATLAS-SUSY-2019-08 [28] (other analyses in the SMODELS database do not give any relevant constraints). Assuming that hadronic searches (vetoing leptons) and leptonic ones (requiring leptons) are independent to good approximation, we can combine all three analyses. The result is shown in Fig. 9. The combined sensitivity increases to $r_{\text{exp}} = 1.67$, while the observed $r$-value moves to $r_{\text{obs}} = 1.32$, compared to $r_{\text{exp}} = 1.52$ and $r_{\text{obs}} = 1.41$ in Fig. 1.

Figure 10 shows an EW-ino scenario with a bino-like $\tilde{\chi}_1^0$ with mass around 140 GeV, and higgsino-like $\tilde{\chi}_{2,3}^0, \tilde{\chi}_1^\pm$ with masses around 555 GeV; the wino-like states have masses of 2.3 TeV. This scenario is not excluded by any of the individual analyses in the SMODELS v2.3.0 database. Concretely, we get the following results:

| Analysis | final state | $r_{\text{exp}}$ | $r_{\text{obs}}$ |
|---|---|---|---|
| CMS-SUS-21-002 | hadronic | 0.81 | 0.63 |
| ATLAS-SUSY-2018-41 | hadronic | 0.63 | 0.92 |
| ATLAS-SUSY-2018-05 | $2\ell$ + jets | 0.46 | 0.68 |
| ATLAS-SUSY-2019-08 | $1\ell + b\bar{b}$ | 0.42 | 0.27 |

Again, the highest sensitivity comes from the hadronic EW-ino searches ATLAS-SUSY-2018-41 and CMS-SUS-21-002, and combining them one reaches $r_{\text{exp}} = 1.06$ and $r_{\text{obs}} = 1.05$. In Fig. 6

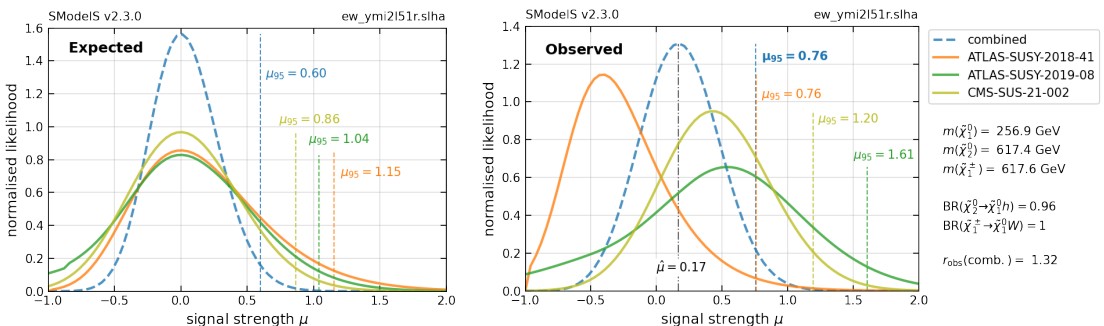

Figure 9: Same as Fig. 1 but including also the ATLAS-SUSY-2019-08 analysis ($1\ell + b\bar{b} + \not{E}_T$ final state) in the combination. The combined sensitivity increases to $r_{\text{exp}} = 1.67$, while the observed $r$-value moves to $r_{\text{obs}} = 1.32$ (compared to $r_{\text{exp}} = 1.52$ and $r_{\text{obs}} = 1.41$ in Fig. 1).

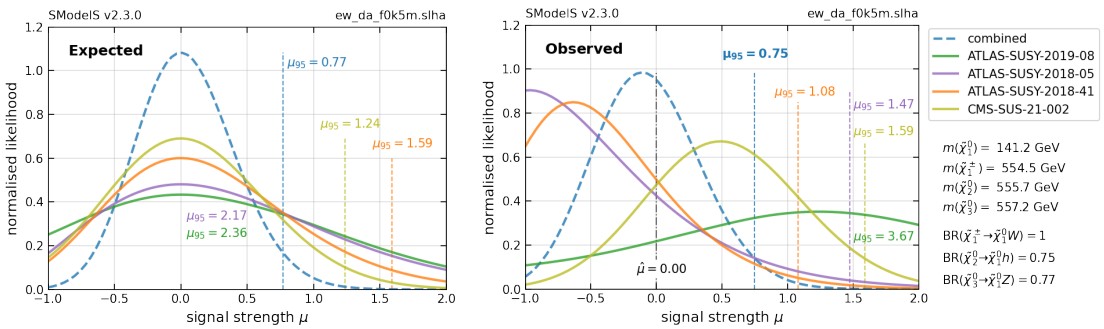

Figure 10: Visualisation of likelihoods for an EW-ino sample point with bino-like $\tilde{\chi}_1^0$ and higgsino-like $\tilde{\chi}_1^{\pm}$, $\tilde{\chi}_{2,3}^0$ with masses $m_{\tilde{\chi}_1^0} = 141$ GeV, $m_{\tilde{\chi}_1^{\pm}, \tilde{\chi}_{2,3}^0} \approx 555$ GeV. See text for details.

in section 4, the point thus appears as newly excluded by the combination. Nonetheless the other two ATLAS analyses are not irrelevant: including them in the combination, the sensitivity increases to $r_{\text{exp}} = 1.30$, and the observed exclusion increases to $r_{\text{obs}} = 1.34$, see Fig. 10.[11] It is interesting to note how the deficits of events in ATLAS-SUSY-2018-41 and ATLAS-SUSY-2018-05 on the one hand, and the small excesses of events in CMS-SUS-21-002 and ATLAS-SUSY-2019-08 on the other hand are averaged out so that in the end $r_{\text{obs}} \approx r_{\text{exp}}$ in the combination.

# D   Auxiliary plots supplementing the results of section 4

In this Appendix we present a number of additional plots which may be of interest to the reader. In particular, these are projections of the results from section 4 in terms of the gaugino and higgsino mass parameters $M_1$, $M_2$ and $\mu$, giving more insights into the mass hierarchies and mixings at play.

Figure 11 shows the scan points excluded by the most sensitive or the most constraining analysis, equivalent to Figs. 3 and 4 in section 4. The three cases considered are:

 – top row: $M_1 < M_2 < \mu$, leading to a bino-like $\tilde{\chi}_1^0$ and wino-like $\tilde{\chi}_2^0$ and $\tilde{\chi}_1^{\pm}$;
 – middle row: $M_1 < \mu < M_2$, leading to a bino-like $\tilde{\chi}_1^0$, higgsino-like $\tilde{\chi}_{2,3}^0$ and $\tilde{\chi}_1^{\pm}$; and
 – bottom row: $\mu < M_1, M_2$, leading to higgsino-like and near mass-degenerate $\tilde{\chi}_{1,2}^0$ and $\tilde{\chi}_1^{\pm}$.

---

[11]The alert reader will notice that $\hat{\mu} = 0$ in Fig. 10, while the value that maximises the likelihood is below zero. The reason is that we do not give any physical meaning to negative signal strengths and therefore limit $\hat{\mu} \geq 0$.

The gaugino/higgsino mixing is further illustrated in Fig. 12, which shows the scan points excluded by the most constraining analysis in the plane of $M_2/M_1$ versus $M_2/\mu$. The points below the dashed line are in the compressed region ($m_{\tilde{\chi}_1^\pm} - m_{\tilde{\chi}_1^0} < 10$ GeV) and correspond to the higgsino LSP scenario. Note that the wino-LSP scenario ($M_2 < M_1, \mu$) is not considered, since it leads to long-lived charginos.

Figure 11: Scan points excluded by the most sensitive analysis (left panels) or the most constraining analysis (right panels) in terms of $M_1$, $M_2$ and $\mu$. Three different hierarchies are considered: $M_1 < M_2 < \mu$ (top row), $M_1 < \mu < M_2$ (middle row) and $\mu < M_1, M_2$ (bottom row).

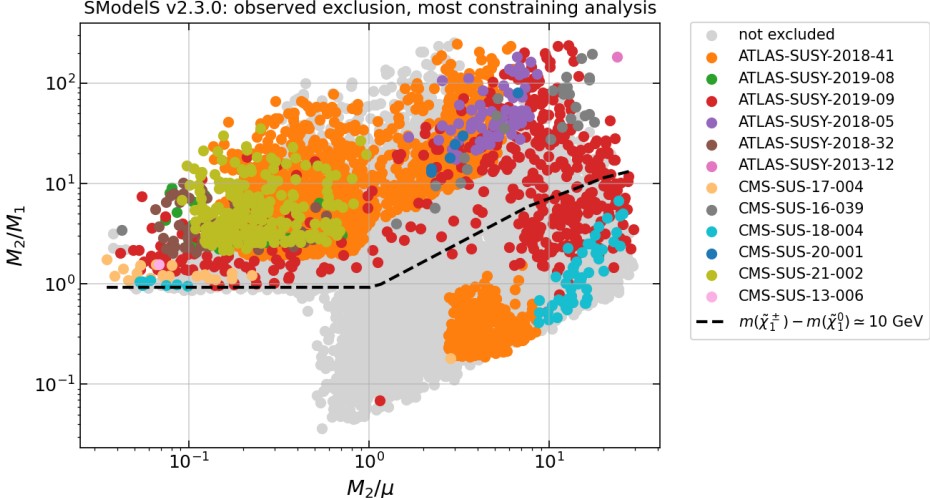

Figure 12: Scan points excluded by the most constraining analysis in the plane of $M_2/M_1$ versus $M_2/\mu$.

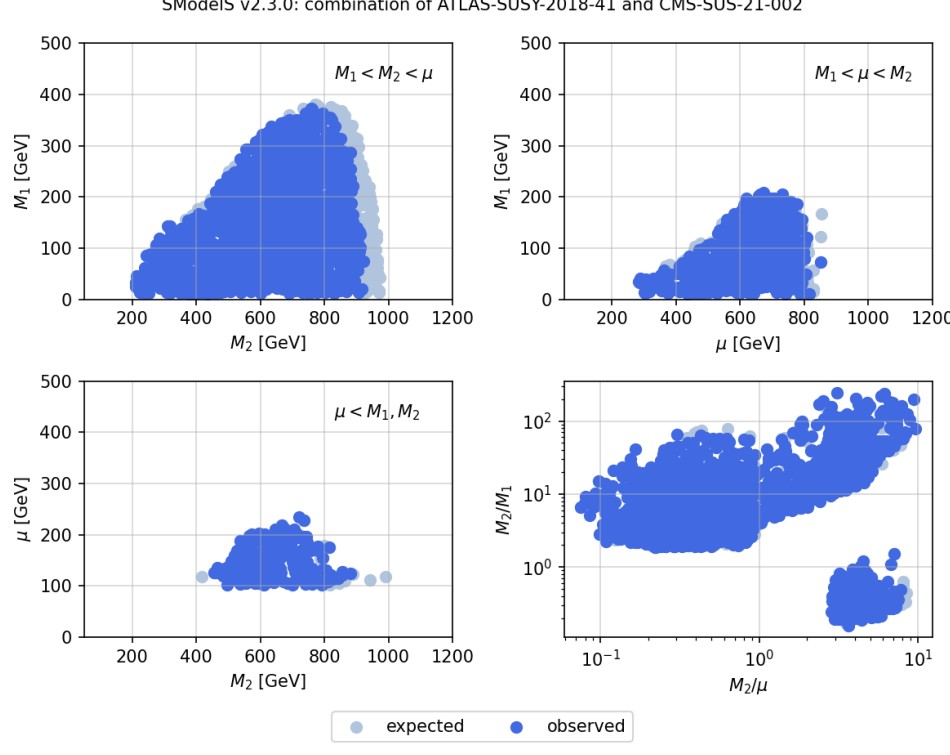

Figure 13: Expected and observed exclusion by the combination of the ATLAS and CMS hadronic EW-ino searches in terms of $M_1$, $M_2$ and $\mu$. Note that, except for a small strip at high $M_2$, the expected and observed exclusions are almost the same.

Last but not least, as a supplement to Figs. 5 and 6, Fig. 13 shows the expected and observed exclusion obtained from the combination of the ATLAS and CMS hadronic EW-ino searches in the $M_1$ vs. $M_2$, $M_1$ vs. $\mu$ and $\mu$ vs. $M_2$ planes for the same three cases as above (top left, top right and bottom left panels, respectively). The forth panel on the bottom right shows the projection of the excluded points onto the $M_2/M_1$ versus $M_2/\mu$ plane.

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
