# Peer review of "SModelS v2.3: enabling global likelihood analyses"

_SciPost Physics, doi:SciPost Phys. 15, 185 (2023)_

## Round 1 · Referee Report · Anonymous (Referee 1) · 2023-8-2

Report

The main purpose of the paper "SModelS v2.3: enabling global likelihood analyses" by Altakach ea. is to report about new features of a public tool for fast reinterpretation of LHC searches, SModelS. It is a valuable contribution that neatly summarises the new features. The paper is well written and the message is clear. An example of a combination analysis can be easily reproduced even for an inexperienced user.

The main new feature is a possibility to combine "orthogonal" LHC searches. Orthogonality would typically mean, according to the paper, combination of ATLAS and CMS results or hadronic and leptonic searches. The decision about validity of combination is left to a user. Perhaps it would be worthwhile to expand the recommendations regarding this point.

The combination feature is illustrated using 2 newly added searches CMS-SUS-21-002 and ATLAS-SUSY-2018-41, which target boosted hadronic decays of W/Z/H + MET . In Figs. 3-5 a result of scan is presented. A problematic thing here are the orange (yellow(khaki?)/blue in Fig. 5) points along m_chargino=m_LSP line. This is very misleading even though there is a comment in the text that these are actually Higgsino and the exclusion is in fact due to heavy winos. I think the problem is that a MSSM-parameters based model is presented in a way simplified model are being shown. Authors should consider presentation of a plot(s) in terms of M1/M2/mu parameters instead.

Regarding the example discussed in Sec. 2, I think it would be interesting to extend discussion with a pair of points marked as blue/red in Fig. 5, e.g. those at ~(700, 200) GeV. Finally, most of the dots are yellowish to me-this is something I see both in print and on screen-but the caption says "khaki" which means dull "greenish"?

Finally, the last sentence of Sec. 4 says "combination of analyses (...) leads to statistically more robust limits". I am not sure how to interpret this statement? In Fig. 5 the number of newly excluded and unexcluded points seem to be rather similar. I do not see this statement justified. Clearly, there is an obvious impact of combined SRs seen in Fig. 2, but this is quite a different effect. While I understand "robustness" argument, the chosen example does not immediately support this statement.

To summarize, the paper provides a quick reference guide to the recent changes in SModelS. The new features are illustrated with an interesting physics example. This warrants publication after minor changes in discussion and presentation of the results, as specified above.

Requested changes

  1. Consider a change in presentation of results of the scan in Figs. 3-5. At least a comment in the caption is necessary. Fix coloring issue in Fig. 5.

  2. Consider adding another benchmark illustrating combination of analyses.

  3. "Robustness" should be elaborated or the statement modified.

---

## Round 1 · Referee Report · Anonymous (Referee 2) · 2023-8-3

Strengths

1- Improvement of the exclusion region of supersymmetric searches by combination of measurements (simplified) 2- Extended coverage of analyses compared to the last publication

Weaknesses

1- Section 2.2: the criteria to declare two analyses "approximately" uncorrelated is not defined. 2- Section 2.2: explanation of the limitations of the approximations should be clarified.

Report

The publication addresses the important task of the combination of experimental analyses in simplified scenarios. The updates of the software package itself as well as the database of useable analyses is more than an incremental step, justifying a new publication.
I recommend publication after the questions have been addressed.

Requested changes

1- Which degree of correlation is accepted in the combination and how is it evaluated? 2- The SRs are quoted as being orthogonal, but is this sufficient as the background estimation in the SR may be based on control regions? Isn't it necessary to also ensure that the CR of analysis A is orthogonal to SR of analysis B et cetera? 3- ULs depend on the statistical and systematic errors. Section 2.2 and equation 6 discuss the combination of orthogonal analyses. Couldn't orthogonal have correlated errors, eg the error on the measurement on the integrated luminosity? How does the combination ensure that such errors are not reduced by combining N analyses?

  • validity: high
  • significance: high
  • originality: high
  • clarity: top
  • formatting: excellent
  • grammar: excellent

Author:  Sabine Kraml  on 2023-08-22  [id 3915]

(in reply to Report 1 on 2023-08-03)

Regarding point 1, "which degree of correlation is accepted in the combination and how is it evaluated":
We are not evaluating the degree of correlation of analyses. SModelS only provides the capability of combining analyses. The working assumption here is that the correlation of systematic uncertainties for combinable analyses are negligible. Which analyses are actually combinable under this assumption has to be defined by the user. This also applies to point 2, that one should also ensure that the CRs of analysis A are orthogonal to the SRs of analysis B, etc. We add a line of caution in the revised manuscript.

Regarding point 3, it is true that orthogonal analyses can still have correlated errors, e.g. the error on the measurement on the integrated luminosity. In general the effect of neglecting this type of uncertainty is considered negligible compared to other approximations in the whole simplified model approach. However, the user still has to apply common sense; e.g., they should not combine N analyses of very limited sensitivity. Again, the tool provides the technical capability, but it is up to the user to use it sensibly.

---

## Round 2 · Referee Report · Anonymous (Referee 1) · 2023-9-4

Report

The authors have appropriately addressed the points raised in the first report. I recommend the paper for publication.

---

## Round 2 · Referee Report · Anonymous (Referee 2) · 2023-9-11

Report

The authors have adequately addressed my comments to the first version of the manscript, therefore I recommend to publish the paper.

---

## Round 2 · Author Response

We thank the referees for their positive assessments and the constructive criticism, which made us improve our paper. We made an effort to properly address all points raised by the referees (and more) and hope that the paper is now ready for publication in SciPost Physics. The changes to the manuscript are listed below.

---

## Round 2 · List of Changes

• As requested by both referees, we have added a statement at the beginning of section 2.2 (page 5) clarifying what we mean by "approximately uncorrelated" analyses. Moreover, we now mention in footnote 4 the possibility of overlaps of signal and control regions in different analyses. This now reads:

"By approximately uncorrelated we mean that signal regions do not overlap and inter-analyses correlations of systematic uncertainties (stemming, e.g., from luminosity measurements) can be neglected. (footnote: Overlaps of SRs of one analysis with the control regions of another analysis in the combination can in principle induce correlations of systematic uncertainties and therefore should also be checked. However, we generally expect the effect to be negligible compared to other uncertainties in SModelS.)"

  • Related to the above, on page 6, 2nd paragraph, we added a remark regarding reference [17] as an example for an approach to explicitly testing (and quantifying) analyses correlations.

However, we do not wish to give explicit recommendations as to the validity of analyses combinations: in the present paper, we present the new functionalities of the new SModelS version, but, as mentioned in the text, it is the responsibility of the user to apply them sensibly. This holds in particular for the question which analyses they treat as combinable.

  • In response to the remark by the 2nd referee, that “Regarding the example discussed in Sec. 2, I think it would be interesting to extend discussion with a pair of points marked as blue/red in Fig. 5, e.g. those at ~(700, 200) GeV.“ we have added the paragraph starting with “We like to stress here …” as next-to-last paragraph in section 2.2 (page 7), as well as two explicit examples in section 4 (new Figs. 7 and 8, and a paragraph discussing them on page 13).

  • In response to the question by the 2nd referee about the “robustness” argument in section 4, we have added a new figure showing the effect of statistically combining the hadronic EW-ino searches from ATLAS and CMS, ATLAS-SUSY-2018-41 and CMS-SUS-21-002 on the expected reach (now Fig. 5) and revised the paragraph starting with “The assessment of the excluded parameter space can be improved …” accordingly. Moreover, we have slightly expanded the last paragraph on page 13 and the concluding paragraph of section 4 (on page 14) to clarify what we mean by more robust constraints.

  • Still in response to the 2nd referee: khaki is the name of the colour in matplotlib; for better readability, we changed “khaki” to “khaki (light yellowish)” in the text and the caption of Fig. 6. The choice of colour gives a good contrast in the plot without being too dominant, so we do not wish to change it.

  • The 2nd referee criticised the presentation of results in section 4 in terms of neutralino vs. chargino mass and suggested that we “consider presentation of a plot(s) in terms of M1/M2/mu parameters instead.” To our mind, the presentation in Figs. 3-6 in terms of neutralino vs. chargino mass is more appropriate for our discussion. But, we have added an appendix (Appendix D) with auxiliary plots that show our results in terms of M1/M2/mu parameters. We hope this is satisfactory to the referee.

Other changes:

  • In addition to the changes triggered by the referees’ comments, we have corrected a number of typos and updated some references. Moreover, we corrected a mistake in the 3rd paragraph on page 6: "In the former case, the observed limit is weaker, in the latter case stronger than the expected limit." (“stronger” and “weaker” were interchanged).

  • We also realised that 13 of our SLHA input files had wrong neutralino2 decays due to a bug in SOFTSUSY; we have taken these 13 points out of the scan data, which leaves us with 18544 scan points instead of 18557 points in version 1. This has no impact on any of the plots or conclusions presented.

  • Our results for SModelS v2.1 and v2.3 w/o SR combination had been done with a sigmacut parameter of 1e-3 fb, but those for v2.3 with SR combination had been done with the default sigmacut=5e-3 value. It turned out that the results in higgsino LSP region are somewhat sensitive to this sigmacut parameter. We have therefore re-run with a sigmacut of 1e-3 fb. This affects the exclusion status of about 100 points mostly with higgsino LSP from SModelS v2.3 with SR combination. All numbers and plots (Figs. 2-6) are updated accordingly. We stress that, apart from the strip at m(chargino1)\approx m(neutralino1), there is no visible effect in the figures.

  • Last but not least, we realised that users might expect the ATLAS-SUSY-2018-16 analysis (the Run 2 search for electroweak production with compressed mass spectra) to be implemented in the SModelS database, though it is not (because it is not compatible with the approximations made in SModelS). We therefore added a clarifying remark in section 3, at the end of the first paragraph of page 8. And we changed “the full suit of available electroweak-ino searches” to “a large variety of electroweak-ino searches” in the abstract to avoid confusion about this point.

---

## Editorial Decision

published